# Single-shot on-chip spectral sensors based on photonic crystal slabs

Zhu Wang[1], Soongyu Yi[1], Ang Chen[1], Ming Zhou [1], Ting Shan Luk[2], Anthony James[2], John Nogan[2], Willard Ross[2], Graham Joe[1], Alireza Shahsafi[1], Ken Xingze Wang[3], Mikhail A. Kats[1] & Zongfu Yu[1]

Miniaturized spectrometers have significant potential for portable applications such as consumer electronics, health care, and manufacturing. These applications demand low cost and high spectral resolution, and are best enabled by single-shot free-space-coupled spectrometers that also have sufficient spatial resolution. Here, we demonstrate an on-chip spectrometer that can satisfy all of these requirements. Our device uses arrays of photodetectors, each of which has a unique responsivity with rich spectral features. These responsivities are created by complex optical interference in photonic-crystal slabs positioned immediately on top of the photodetector pixels. The spectrometer is completely complementary metal–oxide–semiconductor (CMOS) compatible and can be mass produced at low cost.

[1] Department of Electrical and Computer Engineering, University of Wisconsin–Madison, Madison, WI 53705, USA. [2] Center for Integrated Nanotechnologies, Sandia National Laboratories, Albuquerque, NM 87185, USA. [3] Department of Physics, Huazhong University of Science and Technology, Wuhan, Hubei, China. Correspondence and requests for materials should be addressed to Z.Y. (email: zyu54@wisc.edu)

Current use of spectroscopy is still largely confined to laboratories because spectrometers are bulky, expensive, and delicate. There has been tremendous interest in miniaturizing spectrometers to enable a broader range of applications[1]. There are two classes of compact spectrometers: waveguide-coupled[2–10] and free-space coupled[11–21]. The waveguide-coupled spectrometers have limited applications, because they require delicate couplers and do not offer spatial information. On the other hand, free-space coupled devices offer much broader use[11,12,18,22–31] such as imaging. The most important example is the color camera that relies three spectral filters: Red(R), Green(G), Blue(B). To go beyond three spectral bands, Fabry-Perot[13] and plasmonic filters[12,27,29] have been developed. However, these resonant filters have simple Lorentz line shapes and lack the spectral diversity to provide high spectral resolution. It was theoretically shown that random spectral filters can offer high spectral resolution when combined with advanced signal processing methods such as compressive sensing[32]. Recently, a seminal work by Bao and Bawendi[17] experimentally demonstrated a high-resolution spectrometer based on random spectral filters. It showcased a different path other than the resonant filters. The diverse range of spectral features are created by absorption of colloidal quantum dots.

Built upon these progresses, here we developed a scalable method to realize random spectral filters based on photonic crystals (PCs). In contrast to quantum dots where the fabrication could be complicated by the use of non-standard complementary metal–oxide–semiconductor (CMOS) materials and processes, PC slabs can be defined via single exposure photolithography and only require standard CMOS materials. As the spectral response functions are entirely extrinsic and enabled by structures instead of materials' properties, the concept can be applied to any wavelength range by scaling the dimension of PC. They are also extremely compact, with sizes similar to light-sensing pixels in CMOS image sensors. They provide single-shot measurement, which is particularly important for mobile applications.

## Results

**Working principle of PC spectrometers.** The difficulty of reducing spectrometer size arises from the fact that a long propagation path is needed for light of similar wavelengths to accumulate a detectable difference in phase. In typical monochromators or Fourier-transform spectrometers, light only passes through the instrument once or twice. As a result, the optical path length is limited by the physical size of the instrument. One effective way to reduce the spectrometer size is to utilize multiple reflections of light within microstructures. Microcavities have been shown to increase the optical path length to millions of times larger than their physical size. Compact spectrometers have been demonstrated based on photonic-crystal cavities[33], micro-donut[6], and micro-ring resonators[34,35]. However, the path enhancement in microcavities only occurs at selected wavelengths. As a result, many distinct cavities are needed to continuously cover even a small spectral range. The larger the enhancement of the optical path, the narrower the cavity linewidth becomes, which further exacerbates the issue of spectral coverage. Narrow linewidths also reject most of incident light, making such spectrometers particularly susceptible to detector noise.

To overcome issues with narrow-band filters, we propose to miniaturize spectrometers based on PC slabs, which are micrometer-thin dielectric layers with periodic patterning. Light incident from free-space can couple to lateral propagation modes, where the periodic nanostructures allow light to bounce back and forth many times. Unlike microcavities, where the path

enhancement only occurs at resonant frequencies, the effect of path enhancement in PC slabs spreads over a broader spectral range and creates a transmission spectrum with rich spectral features, including sharp peaks due to guided resonances[36,37], broad background variation by Fabry-Perot resonance, and irregular line shapes due to Fano interference[38,39]. To work as a spectrometer, arrays of different PC slabs are fabricated on top of a CMOS imaging sensor. Each PC slab has a different periodicity, lattice constant, and hole sizes. These slabs produce different transmission spectra $T_i(\lambda)$, where $\lambda$ is the free-space wavelength. Thus, the photodetector underneath the $i$th PC slab receives the following signal:

$$S_i = \int I(\lambda) T_i(\lambda) \eta(\lambda) \, d\lambda, \tag{1}$$

where $I(\lambda)$ is the spectrum of incident light and $\eta(\lambda)$ is the spectral responsivity of the photodetector, which can be characterized in experiments (Supplementary Note 1).

Spectrometers can be characterized by the spectral response $T_i(\lambda)$ that they use to sample the incident light. As shown in Fig. 1a, b, conventional spectrometers such as monochromators and Fourier-transform spectrometers use delta-like or sinusoidal functions to sample incident light. However, creating such spectral response in a compact device can be challenging. Instead of relying on these conventional sampling functions, PC slabs, which are extremely compact, rely on a set of complex and variable spectral functions, as shown in Fig. 1c. These functions form a random basis that allows the incident spectrum to be

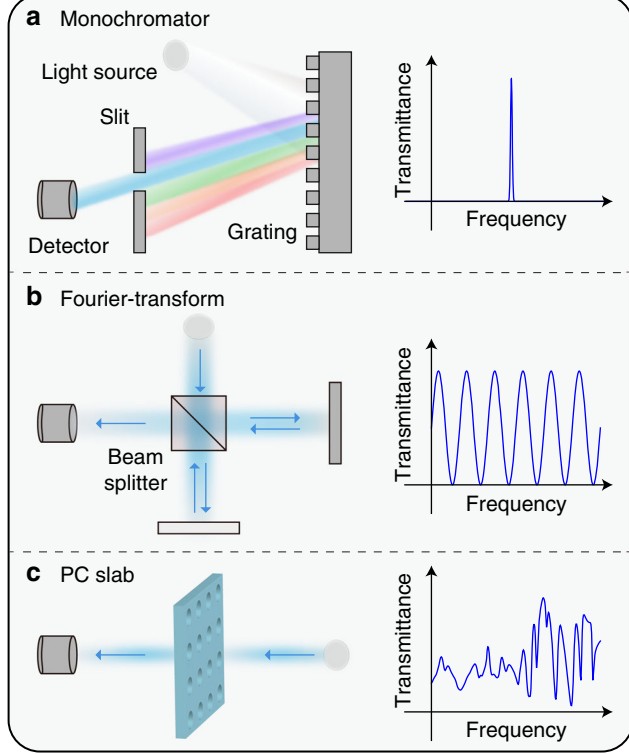

**Fig. 1** Different spectrometers and their spectral responses. **a** The spectral responsivity of grating or microcavity spectrometers has a single sharp peak. The spectrum is measured by point-wise sampling. **b** The spectral responsivity of a Fourier-transform spectrometer is a sinusoidal function. The period varies to perform sampling based on Fourier basis. **c** Photonic-crystal (PC) slabs create complex and variable spectral responsivity depending on the structures. They randomly sample the incident light

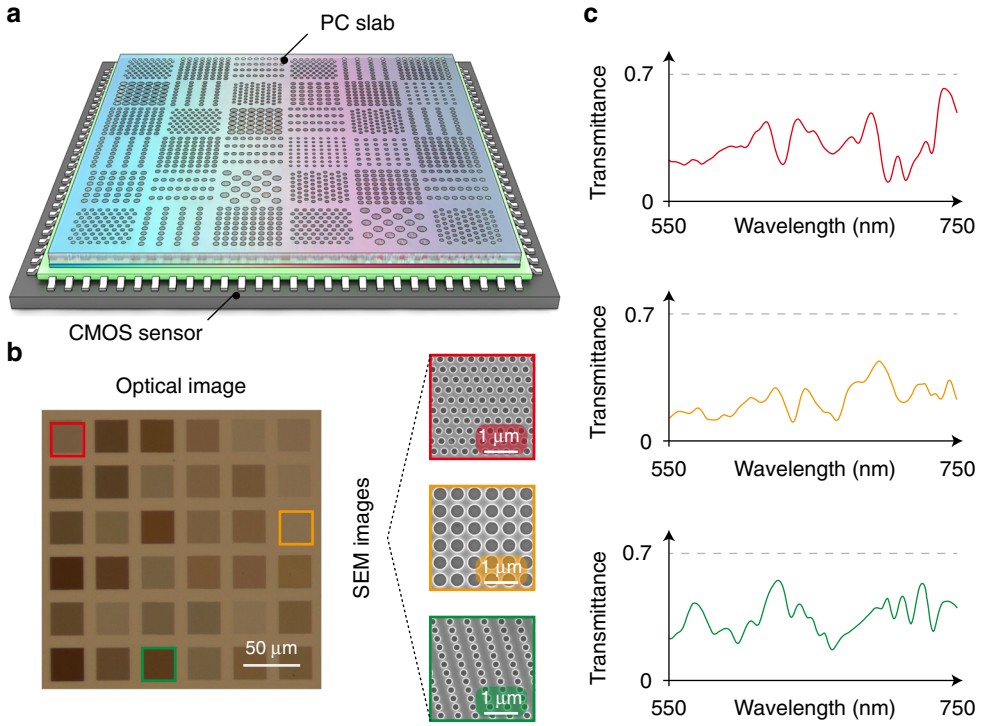

**Fig. 2** Micro-spectrometer based on photonic-crystal (PC) slabs. **a** Schematic of the spectrometer, which consists of an array of PC slabs with different parameters. These slabs are integrated on top of a CMOS sensor array. **b** Optical image of the fabricated 6 × 6 PC structures. Three scanning electron microscopy (SEM) images of selected PC-slab structures marked by red, orange, and green frames, respectively, are shown on the side. **c** Measured transmission spectra $T(\lambda)$ of the three structures in **b**. For each PC slab, the corresponding $T(\lambda)$ is characterized using a monochromator

efficiently recovered using least-square methods[40] or compressive sensing[32].

**Fabrication and characterization of devices.** In this work, we realized a spectrometer, as illustrated in Fig. 2a, operating in the wavelength range of 550 to 750 nm, with a resolution of approximately 1 nm. We used 36 different PC structures. Each PC has a size of 32 × 32 μm, and the entire spectrometer size is 210 × 210 μm (Fig. 2b). The PC structures are chosen to minimize the correlation among different $T_i(\lambda)$, as shown in Supplementary Note 2. Nano-patterns are defined by electron-beam lithography and transferred into a silicon-on-sapphire (SOS) substrate using reactive-ion etching.

After fabrication, we first measured the spectral responsivities of all individual PC slabs $T_i(\lambda)$ using a wavelength-tunable light source incident from the normal direction. Each PC slab covers tens of CMOS pixels (the pixel size is 5.86 μm). Excluding the pixels at the boundaries, which may be only partially covered by the PC slab due to misalignment, we sum signals from all pixels, resulting in a measured responsivity, which is normalized by the CMOS sensor signal without PC slabs. Three representative measurements are shown in Fig. 2c. All $T_i(\lambda)$ were measured at the same time, using an expanded beam that illuminated the entire chip.

The inversion of Eq. 1 to obtain $I(\lambda)$ from $S_i$ is generally an under-determined problem. We used a reconstruction algorithm based on minimizing regularized squares error with nonnegativity constraints (Supplementary Note 1)[41,42].

To test our spectrometer, we first measured light sources with varying intensity and bandwidth. We combined beams from a green LED (Thorlabs, LED570L) and a red LED (Thorlabs, LED630L), as shown Fig. 3a. The combined spectrum was first measured using a commercial monochromator (Spectral Products

DK480), and has two broad peaks centered at 570 and 630 nm (red curve in Fig. 3a). The reconstructed spectrum using our PC spectrometer is shown with blue circles, and agrees very well with the reference. Unlike the monochromator, the PC spectrometer captures a spectrum in a single-shot, within tens of milliseconds when the incident power is tens of μW. By comparison, it took a few seconds for the monochromator to acquire the spectrum. Another multi-color LED (Thorlabs, LEDRY) was tested as shown in Fig. 3b, also resulting in good agreement. We then use the spectrometer to study the metamerism effect, where two different spectra are perceived as the same by human eyes or RGB cameras. The first spectrum is the combination of green and red light in a 3:4 power ratio. The second spectrum has one narrow peak centered at 590 nm. The PC slab spectrometer easily distinguishes the two signals and reconstructs the exact spectral components (Fig. 3c, d). In all of the above measurements, we used the same reconstruction algorithm with identical parameters. The performance can be further improved by tuning the reconstruction algorithm to take advantage of prior knowledge, such as the approximate bandwidth of the measured spectrum.

Next, we tested the resolution of the spectrometer using narrow-band spectra. Here, we further optimized the reconstruction algorithm for narrow-band signals, which involved decreasing the weight of the term that regulates the smoothness of reconstructed spectrum. We measured the emission spectra of HeNe lasers at 594 nm (JDSU, 1137) and 633 nm (Melles Griot, 25-LYR-173-249), respectively (Fig. 4a). The PC spectrometer correctly identified the peak wavelengths, and the results matched well with those obtained using the monochromator. As a further test, we used the spectrometer to measure a series of narrow-band spectra generated by the monochromator. The spectral peaks have bandwidths of about 1.4 nm and wavelengths varying from 550 to 750 nm with a 1 nm step. Figure 4b shows examples of spectra obtained by our spectrometer, which are compared with

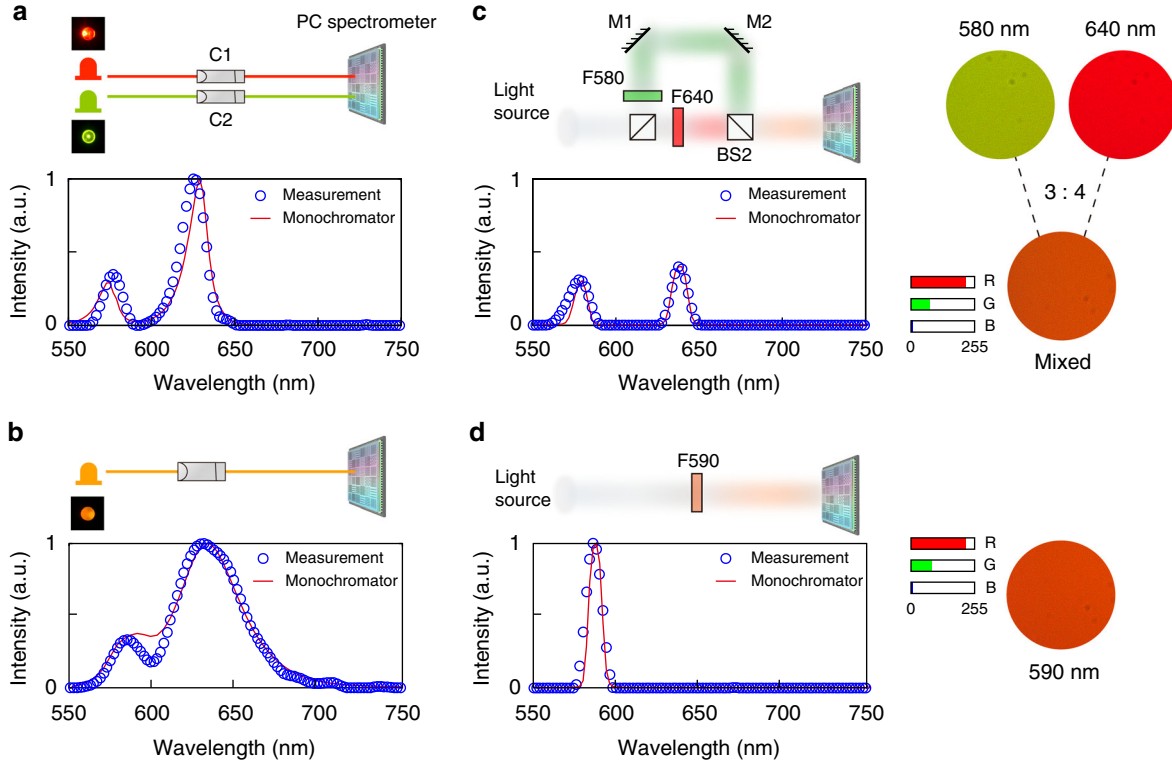

**Fig. 3** Recovery of broadband spectra. Measurements of **a** the emission spectrum of a combination of two LEDs (green and red), and **b** the emission spectrum of a multimode LED (orange/red). C, collimator. **c**, **d** Two different spectra appear as the same RGB color. M, mirror; F, filter; BS, beam splitter. **c** A light beam from a lamp (ASBN-W) is split into two beams, and recombined in a power ratio of 3:4 after passing through two different bandpass filters at 580 ± 5 nm (Thorlabs, FB580) and 640 ± 5 nm (Thorlabs, FB640), respectively. The spectrum of the recombined beam is then analyzed by the PC spectrometer. **d** Measurements for a white light beam (ASBN-W) passing through a bandpass filter at 590 ± 5 nm (Thorlabs, FB590). For all four cases, the measurement results using the PC spectrometer (circled lines) match well with reference spectra obtained by a commercial monochromator (red solid line)

the specification of the monochromator. A total of 201 spectra are measured. The peak positions agree with monochromator specification (Fig. 4c). The shift of the peak position is within ±0.2 nm for 90% of the measurements, and within ±1 nm for all measurements. The bandwidths also match the specification of monochromator well, as shown in Fig. 4d.

The transmission through PC slabs is angle dependent. Thus, our spectrometer requires the incidence angle to be the same for the calibration and measurement. For our device, the performance degrades if the two angles are different by more than 1 degree (See more details in Supplementary Note 3). Practically, this can be overcome by using a collimating aperture to ensure a consistent incident angle. We note that angular sensitivity is not a problem unique to PC slabs, but is also found in other high-resolution spectrometers including grating monochromators and Fourier-transform spectrometers. PC slabs offer additional design flexibility to trade spectral resolution for angular tolerance. For example, we could use fewer periods in the PC slabs. Reduced sizes weaken the effect of the photonic-crystal, and smooth the sharp spectral features, which reduces the spectral resolution but also makes the spectral responsivity less angle dependent.

**Hyperspectral imaging**. Finally, we demonstrated the potential of PC-slab spectrometers for single-shot hyperspectral imaging. The size of each of our spectrometers is around 200 μm on a side. We fabricated 10 × 10 identical spectrometers on an SOS substrate, comprising a total of 3600 PC slabs. Each spectrometer serves as a single spatial pixel. The fabricated device was then attached onto a CMOS sensor chip. As a proof-of-principle demonstration, we used this 10 × 10 pixel imager to acquire a hyperspectral image.

Our target was formed by projecting the numbers "5" and "9" on a piece of paper, as shown in Fig. 5a. The "5" is illuminated with light at a wavelength of 610 ± 5 nm, and the "9" is illuminated at 670 ± 1 nm. These two numbers overlap spatially, and are not easily differentiated using a conventional RGB camera.

A 35-mm lens was used to form an image on the PC-slab sensor chip. We calibrated the spectral responsivity of the device with lens in place. Using a similar reconstruction algorithm as what we used for broadband signals, we obtained a hyperspectral image from which the two numbers can be readily distinguished. Figure 5b shows five images from the reconstructed data from 610 to 690 nm, at increments of 20 nm. Both the spectral and spatial data were successfully retrieved.

The number of pixels of our hyperspectral imager is limited by the throughput of electron-beam lithography, resulting in the low-resolution image in Fig. 5. In practice, more pixels could be added efficiently by using high-throughput photolithography. Hundreds of spatial pixels in each dimension can be realized using the present design. To further increase the spatial resolution, we can also reduce the size of PC slabs with less number of periods used for each PC structure, thus trading spectral resolution for spatial resolution.

## Discussion

In conclusion, we proposed and experimentally demonstrated a spectrometer and a hyperspectral imager based on photonic-crystal (PC) slabs. Traditional hyperspectral imaging techniques often include spatial or spectral scanning, which usually require long acquisition time and are only accurate for stationary scene. Existing non-scanning methods[43–49] often require large optical

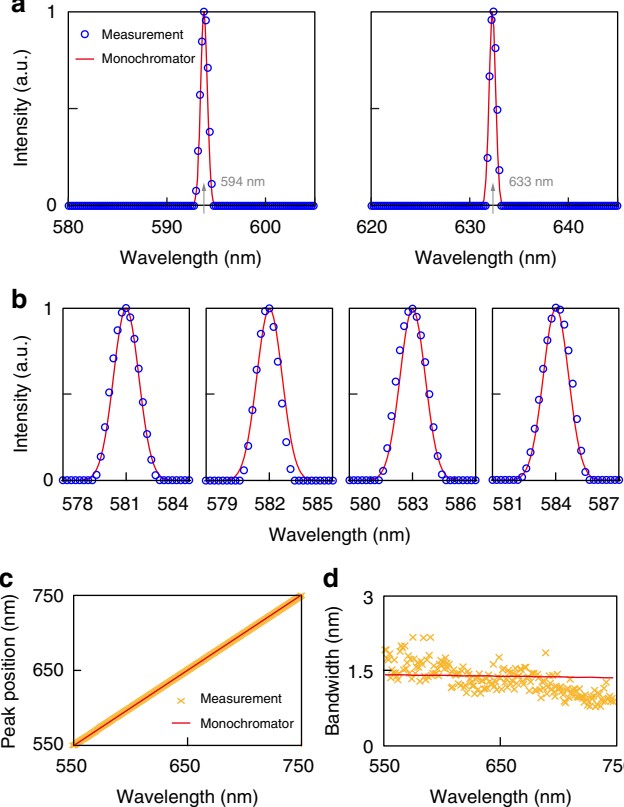

**Fig. 4** Resolution test. **a** Measurements of the emission spectra of two HeNe lasers at 594 and 633 nm, respectively. The reconstructed spectra (circles) match well with the reference spectra (red solid line). The average measured linewidth is around 1 nm. **b** The PC spectrometer is used to measure the spectra of narrow-band light generated by the monochromator at 581, 582, 583, and 584 nm, respectively. The circles are results from the PC spectrometer. The red solid line is the ground-truth reference based on specification of the monochromator. **c** Comparison of 201 measured peak positions (from 550 to 750 nm, with 1 nm separation) with their ground-truth. **d** Comparison of the measured linewidths with the specification of monochromator

components and precision alignment. PC slabs enable non-scanning single-shot imaging method on a compact and reliable CMOS chip, which allows low-cost, portable applications. The performance of demonstrated devices can be readily improved. For example, the spectral resolution can be improved by replacing silicon PC with silicon nitride or silicon carbide, which would create more diverse spectral features due to the lack of light absorption. Spatial resolution and angle tolerance can be improved by using smaller PC slabs.

## Data availability

The data that support the finding of this study are available from the corresponding author upon reasonable request.

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

## Acknowledgements

We would like to thank Dr. Chun-Chieh Chang at Los Alamos National Laboratory for assistance on developing electron-beam lithography recipe. We thank Denise Webb at Sandia National Laboratories for making photomasks for our test patterns. We also thank Liang Zhang at University of Minnesota for help with signal recovery algorithm. The work was partially supported by NSF CAREER Award and DARPA Young Faculty Award (YFA).

## Author contributions

Z.W. performed the numerical simulation, designed the device structure, and performed experimental data analysis. S.Y., T.L., A.J., J.N., Z.W. and W.R. fabricated the device. Z.W., S.Y. and A.C. designed the experimental set-up and performed the measurements. G.J., A.S. and M.K. performed FTIR measurement. All authors contributed to the writing of the manuscript.

## Additional information

**Competing interests:** Ken Xingze Wang and Zongfu Yu hold financial interests in Coherent AI LLC, a company that licensed the technology for commercialization. The remaining authors declare no competing interests.

