## [Peer Review File · Nature Communications]

Reviewers' Comments:

Reviewer #1:

Remarks to the Author:

This is a well written manuscript, describing carefully performed cutting-edge research that is certainly worthy of publication in Nature Comms. The ability to perform hyperspectral imaging with miniaturized spectrometers has a wide range of important applications and is of interest to a wide audience.

My only complaint, and frankly great surprise, is the very little credit they give to the seminal work of Bao & Bawendi, Nature (2015) which is ref. 15 in their paper. To my knowledge, B&B were the first group to demonstrate this approach of employing large arrays of sensors (i.e. QDs in their case) with very different transmission spectra to enable accurate reconstruction of incoming signals. Moreover, when I went back to read the B&B paper, I was struck by the similarity of the two presentations (Wang et al. and B&B) in terms of text and figures (other than the last one in Wang et al.). It's the same idea of using pixels with structured transmission spectra. Wang et al. create theirs using e-beam photolithography, while B&B use QDs with different absorption spectra. Wang et al. have an angle dependence to their response, while B&B do not. Wang et al. claim that B&B are limited in wavelength range, but this is not correct. QDs can be designed to operate from the UV (300nm) to the MWIR, so that's not a real limitation. One important advantage of Wang et al. is that it is much easier to write their structures using lithography on small pixels, while I do not believe there is a QD printing process that can go down to the micron scale reproducibly. Another important advantage of Wang et al. is their ability to control and create very complex spectra by exploiting the flexibility of structural design of photonic crystal slabs.

Again, the work of Wang et al. is very good and deserves publication, but appropriate credit to B&B is warranted.

Reviewer #2:

Remarks to the Author:

The present paper submitted for publication to Nature Communications reports on the fabrication and successful characterisation of a miniaturized spectrometer fabricated on top of and integrated with a CMOS chip. The operation of the spectrometer is based on complex optical interference in photonic crystal slabs. The authors exploit the advantage of photonic crystal slabs to enhance the optical path and this is achieved over a larger range compared to microcavities such as micro-ring or micro-donut resonators. The main characteristics of the device reported herewith are rather clear: it is a free-space, single shot device, with a straightforward fabrication procedure, operating between 550 and 750 nm with a resolution of approximately 1 nm, that can be deteriorated by angles of incidence larger than 1 degree.

What is not clearly presented however and the authors should address upfront in the manuscript, is what are the exquisite and unique features making this work suitable for publication in Nature Communications. The authors start with referencing a rather bulky set of papers (Refs. 1 to 21). If we are considering only several from these references, e.g., refs. 5, 6, 8, 10, each of them reports on "compact spectrometers" with performance that could be considered superior to the device reported here. Of course, many times it is a matter of trade-off among various features, but this needs to be better, clearly and upfront explained for the readership of Nature Communications, that is a broad scientific community, with expertise in various other fields.

Response to reviewer #1:

“This is a well written manuscript, describing carefully performed cutting-edge research that is certainly worthy of publication in Nature Comms. The ability to perform hyperspectral imaging with miniaturized spectrometers has a wide range of important applications and is of interest to a wide audience.”

We would like to thank the reviewer for his/her generous comments, which are very encouraging to us. Thank you.

“My only complaint, and frankly great surprise, is the very little credit they give to the seminal work of Bao & Bawendi, Nature (2015) which is ref. 15 in their paper. To my knowledge, B&B were the first group to demonstrate this approach of employing large arrays of sensors (i.e. QDs in their case) with very different transmission spectra to enable accurate reconstruction of incoming signals. Moreover, when I went back to read the B&B paper, I was struck by the similarity of the two presentations (Wang et al. and B&B) in terms of text and figures (other than the last one in Wang et al.). It’s the same idea of using pixels with structured transmission spectra. Wang et al. create theirs using e-beam photolithography, while B&B use QDs with different absorption spectra. Wang et al. have an angle dependence to their response, while B&B do not. Wang et al. claim that B&B are limited in wavelength range, but this is not correct. QDs can be designed to operate from the UV (300nm) to the MWIR, so that’s not a real limitation. One important advantage of Wang et al. is that it is much easier to write their structures using lithography on small pixels, while I do not believe there is a QD printing process that can go down to the micron scale reproducibly. Another important advantage of Wang et al. is their ability to control and create very complex spectra by exploiting the flexibility of structural design of photonic crystal slabs.”

We lost our track of perspective. The B&B paper is now properly credited, which reads as follows

“Recently, a seminal work by Bao & Bawendi experimentally demonstrated a high-resolution spectrometer based on random spectral filters. It showcased a different path other than the resonant filters. A diverse range of spectral features are created by absorption of colloidal quantum dots. Built upon these progress, here we developed a scalable method to realize random spectral filters based on photonic crystals.”

To help the reviewer understand our initial perspective, we published the theoretical paper in Optics Express in 2014 (Vol. 22, 25608). In writing the paper, we viewed this work as the experimental realization of our theory and design that were published earlier than B&B. But we agree with the reviewer that B&B’s work should be highlighted in the introduction.

“Again, the work of Wang et al. is very good and deserves publication, but appropriate credit to B&B is warranted.”

Response to reviewer #2:

“The present paper submitted for publication to Nature Communications reports on the fabrication and successful characterisation of a miniaturized spectrometer fabricated on top of and integrated with a CMOS chip. The operation of the spectrometer is based on complex optical interference in photonic crystal slabs. The authors exploit the advantage of photonic crystal slabs to enhance the optical path and this is achieved over a larger range compared to microcavities such as micro-ring or micro-donut resonators. The main characteristics of the device reported herewith are rather clear: it is a free-space, single shot device, with a straightforward fabrication procedure, operating between 550 and 750 nm with a resolution of approximately 1 nm, that can be deteriorated by angles of incidence larger than 1 degree.”

We thank the reviewer for carefully reviewing the work and for his/her constructive suggestions.

“What is not clearly presented however and the authors should address upfront in the manuscript, is what are the exquisite and unique features making this work suitable for publication in Nature Communications. The authors start with referencing a rather bulky set of papers (Refs. 1 to 21). If we are considering only several from these references, e.g., refs. 5, 6, 8, 10, each of them reports on compact spectrometers with performance that could be considered superior to the device reported here. Of course, many times it is a matter of trade-off among various features, but this needs to be better, clearly and upfront explained for the readership of Nature Communications, that is a broad scientific community, with expertise in various other fields.”

We thank the reviewer for pointing out this. When we read again the paper, we feel that the significance is indeed buried for a broad readership. The short answer is that it is the first demonstration of a compact spectrometer that can offer very high-resolution spectrometer (~ 1 nm resolution) and imaging capability, which paves the way toward the wide use in consumer electronics.

Compared to the large body of literature cited here, most compact spectrometers including 5, 6, 8, 10 are not free space coupled, and thus cannot provide any spatial information. Other works that do offer free-space coupling usually lack the high spectral resolution and involve complex fabrication, because of the difficulty in creating a diverse range of different spectral filters. The key innovation of this work is the method to design such filters that offer a high spectral resolution and can be fabricated easily.

We have rewritten the introduction as follows:

“Current use of spectroscopy is still largely confined to laboratories because spectrometers are bulky, expensive, and delicate. There has been tremendous interest in miniaturizing spectrometers to enable a broader range of applications¹. There are two classes of compact spectrometers: waveguide coupled^{2–10} and free space coupled^{11–21}. The waveguide-coupled spectrometers have limited applications, because they require delicate couplers and do not offer spatial information. On the other hand, free-space coupling offers much broader use^{4,11,12,17,22–31} such as imaging. The most important example is the color camera that relies three spectral filters: Red(R), Green(G), Blue(B). To go beyond 3 spectral bands, Fabry-Perot¹¹ and plasmonic filters^{12,27,29} have been developed. However, these resonant filters have simple Lorentz line shapes and lack the spectral diversity to provide high spectral resolution. It was theoretically shown that random spectral filters can offer high spectral

resolution when combined with advanced signal processing methods such as compressive sensing³². Recently, a seminal work by Bao & Bawendi experimentally demonstrated a high-resolution spectrometer based on random spectral filters¹⁶. It showcased a different path other than the resonant filters. The diverse range of spectral features are created by absorption of colloidal quantum dots. Built upon these progresses, here we developed a scalable method to realize random spectral filters based on photonic crystals (PCs). In contrast to quantum dots where the fabrication could be complicated by the use of non-standard CMOS materials and processes, PC slabs can be defined via single exposure photolithography and only require standard CMOS materials. Because the spectral response functions are entirely extrinsic and enabled by structures instead of materials properties, the concept can be applied to any wavelength range by scaling the dimension of PC. They are also extremely compact, with sizes similar to light-sensing pixels in CMOS image sensors. They provide single-shot measurement, which is particularly important for mobile applications.”

Reviewers' Comments:

Reviewer #1:

Remarks to the Author:

The paper is now acceptable for publication.

Reviewer #2:

Remarks to the Author:

I think that my previous comments have been addressed properly and from my point of view the paper can indeed be accepted for publication in Nature Comm.